# DUAL TRAINING OF ENERGY-BASED MODELS WITH OVERPARAMETRIZED NEURAL NETWORKS

## ABSTRACT

Energy-based models (EBMs) are generative models that are usually trained via maximum likelihood estimation. This approach becomes challenging in generic situations where the trained energy is nonconvex, due to the need to sample the Gibbs distribution associated with this energy. Using general Fenchel duality results, we derive variational principles dual to maximum likelihood EBMs with shallow overparametrized neural network energies, both in the active (aka feature-learning) and lazy regimes. In the active regime, this dual formulation leads to a training algorithm in which one updates concurrently the particles in the sample space and the neurons in the parameter space of the energy at a faster rate. We also consider a variant of this algorithm in which the particles are sometimes restarted at random samples drawn from the data set, and show that performing these restarts at every iteration step corresponds to score matching training. Using intermediate parameter setups in our dual algorithm thereby gives a way to interpolate between maximum likelihood and score matching training. These results are illustrated in simple numerical experiments.

## 1 INTRODUCTION

Energy-based models (EBMs) are explicit generative models which consider Gibbs measures defined through an *energy function $f$*, with a probability density proportional to $\exp(-\beta f(x))$, where $\beta$ is the inverse temperature. Such models originate in statistical physics (Gibbs, 2010; Ruelle, 1969), and have become a fundamental modeling tool in statistics and machine learning (Wainwright & Jordan, 2008; Ranzato et al., 2007; LeCun et al., 2006; Du & Mordatch, 2019; Song & Kingma, 2021). Given data samples from a target distribution, the learning algorithms for EBMs attempt to estimate an energy function $f$ to model the samples density. The resulting learned model can then be used to obtain new samples, typically through Markov Chain Monte Carlo (MCMC) techniques.

The standard method to train EBMs is maximum likelihood estimation, i.e. the learned energy is the one maximizing the likelihood of the target samples, within a certain function class. One generic approach for this is to use gradient descent, where gradients may be approximated using MCMC samples from the trained model. However, this is computationally difficult for highly non-convex trained energies, due to 'metastability', ie the presence of large basins in the energy landscape that trap trajectories for potentially exponential time. This has motivated a myriad of alternative losses to learn EBM energies, such as the popular score matching; see (Song & Kingma, 2021) for a review. All in all, such weaker losses come at the expense of a loss of statistical power, which motivates exploring computationally efficient methods for EBM maximum-likelihood estimation.

EBMs also have structural connections with maximum entropy (maxent) models, which have been studied for decades through Fenchel duality. Dai et al. (2019b) was the first work to leverage similar duality arguments for maximum likelihood EBM training. However, their analysis is restricted to energies lying in RKHS balls (i.e. non-parametric linear models). Despite the appealing optimization properties of RKHS, these spaces of functions typically only contain very smooth functions when the dimension is large (Berlinet & Thomas-Agnan, 2004). A recent line of work—originating in supervised learning—has considered an alternative based on shallow neural networks (Bach, 2017),

which admit a linear representation in terms of a measure over its parameters and are able to adapt to hidden low-dimensional structures in the data. The statistical benefits of the obtained $\mathcal{F}_1$ or *Barron* spaces have recently been studied in the context of shallow EBMs by Domingo-Enrich et al. (2021), who show that they may outperform the RKHS models.

In this work, we focus instead on the computational aspects of training such shallow EBMs. Relying on infinite-dimensional Fenchel duality results for the KL-regularized $L^2$ and $L^\infty$ regression problems over probability measures (App. B), we recast the maximum likelihood training of $\mathcal{F}_1$-EBMs into a min-max problem over measures, and derive a gradient descent-ascent algorithm (Alg. 1) in the associated metric spaces, based on the Wasserstein distance. Crucially, these schemes, defined over idealised parametrisations requiring infinite number of neurons, admit a finite-particle approximation, as in regression or classification. When viewed in terms of particle systems, the dynamics evolve two interacting populations simultaneously: one over the neuron parameters, and the other over the data space (Sec. 3). Moreover, our proposed algorithm naturally interpolates between the primal and dual formualtions, thanks to the relative time-scale between the minimization and maximization steps, and is even able to interpolate between MLE and Score Matching. This dual algorithm is evaluated experimentally in Sec. 5 in a well-calibrated high-dimensional teacher-student environment, which allows us to assess our models against the ground-truth, and test the effect of input dimension. Our experiments confirm that the dual algorithm converges significatively faster than the primal one, suggesting that dual updates might bypass metastability despite high-dimensional and non-convex energy landscapes.

**Related work.** Our work is based on general Fenchel duality results (App. B) that may be useful in applications beyond the main focus of this paper (see App. D). These theorems are a generalization of results stated in the compact case in Domingo-Enrich et al. (2021) in their Appendix D. Similar duality results have been studied extensively in the area of maximum entropy (maxent) models (reviewed in Ch. 12 of Mohri et al. (2012)). The first maxent duality principle was due to Jaynes (1957). Maxent models have been applied since the 1990s in natural language processing and in species habitat modeling among others, and studied theoretically especially since the 2000s (Altun & Smola, 2006; Dudík et al., 2007).

Recently Dai et al. (2019a) leveraged duality arguments in the context of maximum likelihood EBMs, although in a form different from ours. Their duality result works in the more restrictive setting of "lazy" energies lying in RKHS balls and probability measures with $L^2$ densities, and they derive it directly from a general theorem that works for reflexive Banach spaces (Ekeland & Temam (1999), Ch. 6, Thm. 2.1). Our Fenchel duality results, which work for Borel probability measures and feature-learning ($\mathcal{F}_1$) energies, are more general because we must rely on measure spaces, which are non-reflexive Banach spaces. Their algorithm is also different: they do not evolve generated samples, but rather use a transport parametrization of the energy. Dai et al. (2019b) expand the work (Dai et al., 2019a) combining it with Hamiltonian Monte Carlo.

A precursor of modern machine learning EBMs were restricted Boltzmann machines (RBMs), first trained via contrastive divergence or CD (Hinton, 2002) - which estimates the gradient of the log-likelihood via approximate MCMC samples of the trained model. It later led to maximum likelihood training of EBMs (see e.g. Xie et al. (2016; 2017); Du & Mordatch (2019) among many others). A popular variant of CD is persistent contrastive divergence or PCD (Tieleman, 2008; Tieleman & Hinton, 2009), in which the MCMC samples are evolved and reused over gradient computations to be progressively equilibrated. Drawing a comparison with our work, our dual $\mathcal{F}_1$-EBM training algorithm resembles PCD in that both evolve a set of samples over training iterations.

A vast array of EBM losses alternative to maximum likelihood have been developed in recent years (Song & Kingma, 2021) with the goal of avoiding the MCMC procedure, which may be costly for non-convex densities. Some successful ones are score matching (Hyvärinen, 2005) and related methods such as denoising score matching (Vincent, 2011). Building on these, recent works have achieved state of the art image generation (Song & Ermon, 2019; 2020; Ho et al., 2020; Song et al., 2021). We derive the score matching algorithm for $\mathcal{F}_1$ energies and show a continuum of algorithms interpolating between dual maximum likelihood and score matching training.

Finally, our work (in particular App. C) also has links with maximum mean discrepancy (MMD) flows. MMDs are probability metrics that were first introduced in (Gretton et al., 2007; 2012) for kernel two-sample tests, and that have enjoyed ample success with the advent of deep-learning-based generative modeling as discriminating metrics (Li et al., 2015; Dziugaite et al., 2015; Li et al., 2017). Among the MMD literature, the closest work to ours is (Arbel et al., 2019), which study theoretically the convergence of unregularized MMD gradient flow (our equation (26) with $\tilde{\beta}^{-1} = 0$). In their experiments, they observe that noisy updates ($\tilde{\beta}^{-1} > 0$) are needed for good generalization. Our work shows that their algorithm is exactly training maximum likelihood EBMs energies in an RKHS ball of radius that depends on the noise level (see App. C).

## 2 BACKGROUND AND SETUP

In this section, we provide preliminary background on the functional spaces associated to over-parametrized two-layer networks, and on EBMs and their training losses.

**Notation.** If $V$ is a normed vector space, we use $\mathcal{B}_V(\beta)$ to denote the closed ball of $V$ of radius $\beta$, and $\mathcal{B}_V := \mathcal{B}_V(1)$ for the unit ball. If $K$ denotes a subset of the Euclidean space, $\mathcal{P}(K)$ is the set of Borel probability measures, $\mathcal{M}(K)$ is the space of Radon (i.e. signed and finite) measures, and $\mathcal{M}^+(K)$ is the set of non-negative Radon measures. If $\gamma$ is a Radon measure over $K$, then $\|\gamma\|_{\mathrm{TV}} = \int_K d|\gamma|$ is the total variation (TV) norm of $\gamma$, which turns $\mathcal{M}(K)$ into a Banach space. Throughout the paper, and unless otherwise specified, $\sigma : \mathbb{R} \to \mathbb{R}$ denotes a generic non-linear activation function. We use $(\cdot)_+ : \mathbb{R} \to \mathbb{R}$ to denote the ReLU activation, defined as $(z)_+ = \max\{z, 0\}$. We use $\tau$ to denote a fixed base probability measure, possibly used with a subindex to specify the space it is defined over. We use $\mathbb{S}^d \subseteq \mathbb{R}^{d+1}$ for the $d$-dimensional hypersphere and $\log$ for the natural logarithm. We denote the Lebesgue measure by $\lambda$. Given two probability measures $\nu, \nu' \in \mathcal{P}(K)$, $D_{\mathrm{KL}}(\nu\|\nu') = \int_K \log \frac{d\nu}{d\nu'} d\nu$ denotes the KL divergence from $\nu'$ to $\nu$ and $H(\nu, \nu') = -\int_K \log(\frac{d\nu'}{d\tau}) d\nu$ is the cross-entropy.

### 2.1 OVERPARAMETRIZED TWO-LAYER NEURAL NETWORK SPACES

In this work, we will focus on dense function approximation classes generated by overparametrized shallow neural networks. One can distinguish two canonical models, depending on the asymptotic scaling regime. For further background on these regimes, we refer the reader to Chizat et al. (2019).

**Kernel regime.** Let $\mathcal{X} \subseteq \mathbb{R}^{d_1}$, $\Theta \subseteq \mathbb{R}^{d_2}$, $\varphi : \mathcal{X} \times \Theta \to \mathbb{R}$, and $\tau_\Theta$ be a fixed base probability measure over $\Theta$. We define $\mathcal{F}_2$ as the reproducing kernel Hilbert space (RKHS) of functions $f : \mathcal{X} \to \mathbb{R}$ such that for some $h \in L^2(\Theta, \tau_\Theta)$, we have that, for all $x \in \mathcal{X}$, $f(x) = \int_\Theta \varphi(x, \theta) h(\theta) d\tau_\Theta(\theta)$. The RKHS norm of $\mathcal{F}_2$ is defined as $\|f\|_{\mathcal{F}_2} = \inf \left\{ \|h\|_{L^2(\Theta)} \mid f(\cdot) = \int_\Theta \varphi(\cdot, \theta) h(\theta) d\tau_\Theta(\theta) \right\}$ where $\|h\|_{L^2(\Theta)}^2 := \int_\Theta |h(\theta)|^2 d\tau_\Theta(\theta)$ (c.f. Bach (2017)). As an RKHS, the kernel of $\mathcal{F}_2$ is

$$k(x, y) = \int_\Theta \varphi(x, \theta) \varphi(x, \theta) d\tau_\Theta(\theta). \tag{1}$$

**Feature-learning regime.** Set $\mathcal{X}$, $\Theta$ and $\varphi$ as in the previous paragraph, and define $\mathcal{F}_1$ as the Banach space of functions $f : \mathcal{X} \to \mathbb{R}$ such that, for some Radon measure $\gamma \in \mathcal{M}(\Theta)$, for all $x \in \mathcal{X}$ we have $f(x) = \int_\Theta \varphi(x, \theta) d\gamma(\theta)$. We define the norm of $\mathcal{F}_1$ as $\|f\|_{\mathcal{F}_1} = \inf \left\{ \|\gamma\|_{\mathrm{TV}} \mid f(\cdot) = \int_\Theta \varphi(\cdot, \theta) d\gamma(\theta) \right\}$. This construction was introduced by Bach (2017), who first used the notation $\mathcal{F}_1$ and focused in particular on the case $\mathcal{X} \subseteq \mathbb{R}^d$, $\Theta = \mathbb{S}^d$ and $\varphi(x, \theta) = \mathrm{ReLu}^k(\langle(x, 1), \theta\rangle)$ for some $k \in \mathbb{Z}_+$. This space is also known by the name of Barron space (E et al., 2019; E & Wojtowytsch, 2020) in reference to the classic work (Barron, 1993).

Remark that since $\|h\|_{L^1(\Theta)} = \int_\Theta |h(\theta)| d\tau_\Theta(\theta) \le (\int_\Theta |h(\theta)|^2 d\tau_\Theta(\theta))^{1/2} = \|h\|_{L^2(\Theta)}$ by the Cauchy-Schwarz inequality, we have $\mathcal{F}_2 \subset \mathcal{F}_1$: in particular finite-width neural networks belong

to $\mathcal{F}_1$ but not to $\mathcal{F}_2$ (Bach, 2017). The TV norm in $\mathcal{F}_1$ acts as a sparsity-promoting penalty, which encourages the selection of few well-chosen neurons and may lead to favorable adaptivity properties when the target has a low-dimensional structure.

## 2.2 EBMs AND TRAINING LOSSES

Consider a measurable set $\mathcal{X} \subseteq \mathbb{R}^{d_1}$ with a fixed base probability measure $\tau_{\mathcal{X}} \in \mathcal{P}(\mathcal{X})$. If $\mathcal{F}$ is a class of functions (or energies) mapping $\mathcal{X}$ to $\mathbb{R}$, for any $f \in \mathcal{F}$ we can define the probability measure $\nu_f$ as a Gibbs measure with density:

$$\frac{d\nu_{\beta f}}{d\tau_{\mathcal{X}}}(x) := Z_{\beta f}^{-1} e^{-\beta f(x)} \qquad \text{with} \qquad Z_{\beta f} := \int_{\mathcal{X}} e^{-\beta f(y)} d\tau_{\mathcal{X}}(y) ,$$

where $d\nu_{\beta f}/d\tau_{\mathcal{X}}$ is the Radon-Nikodym derivative of $\nu_{\beta f}$ and $Z_{\beta f}$ is the partition function. The parameter $\beta > 0$ is the inverse temperature. We could merge $\beta$ into $\mathcal{F}$ by considering the function class $\{\beta f | f \in \mathcal{F}\}$, but we have decided to keep them separate to showcase the dependency on $\beta$. Gibbs measures are the cornerstone of statistical physics since the seminal works of Boltzmann and Gibbs. Beyond their widespread use across computational sciences, they have also found their application in Machine Learning, by the name of *energy-based models* (EBMs), where the energy function is parametrised using e.g. a neural network.

We denote by $\mathcal{F}_1$-EBMs the energy-based models for which the energy class $\mathcal{F}$ is the unit ball $\mathcal{B}_{\mathcal{F}_1}(1)$ of $\mathcal{F}_1$. Notice that the class $\{\beta f | f \in \mathcal{F}\}$ is equal to the ball $\mathcal{B}_{\mathcal{F}_1}(\beta)$. Such models may be regarded as abstractions of more complex deep EBMs, in that they incorporate feature learning, and they were first studied by Domingo-Enrich et al. (2021), which provide statistical guarantees. They are to be contrasted with $\mathcal{F}_2$-EBMs, for which $\mathcal{F}$ is the unit ball $\mathcal{B}_{\mathcal{F}_2}(1)$. $\mathcal{F}_2$-EBMs, which we study in App. C, have fixed features and showed worse statistical performance in experiments (Domingo-Enrich et al., 2021).

Given samples $\{x_i\}_{i=1}^n$ from a target measure $\nu_p$, training an EBM consists in selecting the best $\nu_{\beta f}$ with energy $f \in \mathcal{F}$ according to a given criterion. Two such criteria are relevant in this paper.

**Maximum likelihood.** The maximum likelihood estimator (MLE) is defined as $\hat{f} = \arg\max_{f \in \mathcal{F}} \prod_{i=1}^n \frac{d\nu_{\beta f}}{d\tau_{\mathcal{X}}}(x_i)$, or, equivalently, as the minimizer of the cross-entropy $H(\nu_n, \nu_{\beta f})$ with the empirical measure $\nu_n = \frac{1}{n}\sum_{i=1}^n \delta_{x_i}$:

$$\hat{f} = \underset{f \in \mathcal{F}}{\arg\min} -\frac{1}{n}\sum_{i=1}^n \log\left(\frac{d\nu_{\beta f}}{d\tau_{\mathcal{X}}}(x_i)\right) = \underset{f \in \mathcal{F}}{\arg\min} \frac{1}{n}\sum_{i=1}^n f(x_i) + \beta^{-1}\log Z_{\beta f}. \qquad (2)$$

The estimated distribution is then simply given by $d\nu_{\beta\hat{f}} = Z_{\beta\hat{f}}^{-1} e^{-\beta\hat{f}} d\tau_{\mathcal{X}}$. By observing that $D_{\mathrm{KL}}(\nu \| \nu') = H(\nu, \nu') - H(\nu)$, where $H(\nu) := H(\nu, \nu)$ is the entropy, minimizing the cross-entropy is equivalent to minimizing the KL divergence when the latter is finite. However, such equivalence is only well-defined in the *population* setting where the empirical measure $\nu_n$ is replaced by its population counterpart $\nu_p$. An appropriate choice of function class $\mathcal{F}$ induces a regularization that prevents the learned Gibbs measure to overfit to the empirical data measure, and presumably approximate $\nu_p$ instead. The MLE enjoys strong statistical properties (Wainwright & Jordan, 2008; Wainwright, 2019), as well as a powerful variational principle as soon as one considers convex function classes (see App. A), but its notorious computational challenges (see Related works section) have motivated alternative approximation metrics to be used to learn EBMs.

Since an arbitrary element $f$ of $\mathcal{F}_1$ can be expressed as $f(x) = \int_{\Theta} \varphi(x, \theta) \, d\gamma(\theta)$, with $\|f\|_{\mathcal{F}_1}$ equal to the infimum of $\|\gamma\|_{\mathrm{TV}}$ for all such $\gamma$, the maximum likelihood function for $\mathcal{F} = \mathcal{B}_{\mathcal{F}_1}(1)$ the problem (2) can be restated as $f_{\mathrm{MLE}} = \int_{\Omega} \varphi(\cdot, \theta) d\gamma_{\mathrm{MLE}}(\theta)$ where

$$\gamma_{\mathrm{MLE}} = \underset{\substack{\gamma \in \mathcal{M}(\Theta) \\ \|\gamma\|_{\mathrm{TV}} \leq 1}}{\arg\min} \frac{1}{n}\sum_{i=1}^n \int_{\Theta} \varphi(x_i, \theta) \, d\gamma(\theta) + \frac{1}{\beta}\log\left(\int_{\mathcal{X}} \exp\left(-\beta \int_{\Theta} \varphi(x, \theta) \, d\gamma(\theta)\right) d\tau_{\mathcal{X}}(x)\right) \quad (3)$$

**Score matching.** An important instance of such weaker metrics is given by Score Matching (SM). The SM metric between two absolutely continuous probability measures $\nu, \nu'$ is defined as $\text{SM}(\nu, \nu') = \int_{\mathcal{X}} |\nabla \log \frac{d\nu}{d\tau_{\mathcal{X}}}(x) - \nabla \log \frac{d\nu'}{d\tau_{\mathcal{X}}}(x)|^2 \, d\nu(x)$. The SM metric is known in information theory as the relative Fisher information. Note that this metric cannot be trivially extended to the case $\nu = \nu_n$, because empirical measures do not have a density with respect to $\tau_{\mathcal{X}}$. To get around this difficulty, note that, if the target measure $\nu_p$ is absolutely continuous with respect to $\tau_{\mathcal{X}}$ and we denote by $f_p(x) = -\beta^{-1} \log \frac{d\nu_p}{d\tau_{\mathcal{X}}}(x)$ its energy, learning an EBM with function class $\mathcal{F}$ under the population loss corresponds to solving $\hat{f} = \arg\min_{f \in \mathcal{F}} \int_{\mathcal{X}} |\nabla f(x) - \nabla f_p(x)|^2 \, d\nu_p(x)$. The insight from Hyvärinen (2005) is that under regularity conditions on $f_p$, via integration by parts we then have $\int_{\mathcal{X}} |\nabla f - \nabla f_p|^2 \, d\nu_p = \mathbb{E}_{\{x_i\}_{i=1}^n} L(f, \nu_n) + C$, where $\mathbb{E}_{\{x_i\}_{i=1}^n}$ denotes expectation over the data set, $C$ is a constant in $f$ which is therefore irrelevant, and

$$L(f, \nu_n) = \frac{1}{n} \sum_{i=1}^n \beta^{-1} \Delta f(x_i) + \frac{1}{2} |\nabla f(x_i)|^2.$$

In practice, we train an EBM via score matching by solving $\hat{f} = \arg\min_{f \in \mathcal{F}} L(f, \nu_n)$. Score matching is computationally more tractable than maximum likelihood and performs well in practice (see Related works section). Statistically, its main drawback is that the SM metric is weaker than the KL divergence, and may fail to distinguish distributions in some instances.

The following proposition, proved in App. G, provides the expression for the loss $L$ and the resulting score matching problem for $\mathcal{F}_1$-EBMs.

**Proposition 1.** *Suppose that $\mathcal{X} \subseteq \mathbb{R}^{d_1}$ is a manifold without boundaries. Assume that $\int_{\mathcal{X}} |\nabla_x \varphi(x, \theta) \cdot \nabla \frac{d\nu_p}{d\tau_{\mathcal{X}}}(x)| \, d\tau_{\mathcal{X}}(x)$ is upper-bounded by some constant $K$ for all $\theta \in \Theta$. Assume also that $\sup_{\theta \in \Theta} \|\nabla_x \varphi(x, \theta)\| < \eta(x)$ and that $\int_{\mathcal{X}} |\eta(x)|^2 \, d\nu_p(x) < \infty$. The optimization problem to train EBMs under the score matching loss over the ball $\mathcal{B}_{\mathcal{F}_1}(1)$ gives $f_{\text{SM}} = \int_{\Omega} \varphi(\cdot, \theta) d\gamma_{\text{SM}}(\theta)$ where*

$$\gamma_{\text{SM}} = \underset{\substack{\gamma \in \mathcal{M}(\Theta) \\ \|\gamma\|_{TV} \leq 1}}{\arg\min} \int_{\Theta} \int_{\mathcal{X}} \left( \frac{1}{2} \nabla_x \varphi(x, \theta) \cdot \nabla_x \int_{\Theta} \varphi(x, \theta') \, d\gamma(\theta') - \beta^{-1} \Delta_x \varphi(x, \theta) \right) d\nu_n(x) d\gamma(\theta) \quad (4)$$

## 3 DUAL $\mathcal{F}_1$-EBM TRAINING VIA MAXIMUM LIKELIHOOD

As a corollary of the duality result from Subsec. B.2, we derive an alternative objective for $\mathcal{F}_1$-EBMs trained via maximum likelihood, the original objective being (3) and we develop an algorithm to solve this alternative problem. To this end, we make:

**Assumption 1.** *Let $\varphi : \mathcal{X} \times \Theta \to \mathbb{R}$ be a continuous function such that either $\mathcal{X}$ is compact or (i) for any fixed $\theta \in \Theta$, $\varphi(x, \theta) \leq \xi(x)$ for some strictly positive $\xi : \mathcal{X} \to \mathbb{R}$, and (ii) $\xi(x) + \log(\xi(x)) = o\left( -\log\left(\frac{d\tau_{\mathcal{X}}}{d\lambda}(x)\right) - (d_1 + \epsilon) \log \|x\|_2 \right)$ as $\|x\|_2 \to +\infty$ for some $\epsilon > 0$.*

In particular, this assumption holds for ReLU network energies when setting $\mathcal{X} = \mathbb{R}^{d_1}$, $\Theta = \mathbb{R}^{d_1+1}$, $\varphi(x, \theta) = \sigma(\langle (x, 1), \theta \rangle) / \|\theta\|$ and $\tau_{\mathcal{X}}$ Gaussian (and in many other settings).

**Theorem 1.** *Under Assumption 1, the problem (3) is the Fenchel dual of*

$$\min_{\nu \in \mathcal{P}(\mathcal{X})} \max_{\substack{\gamma \in \mathcal{M}(\Theta), \\ \|\gamma\|_{TV} \leq 1}} \beta^{-1} D_{KL}(\nu \| \tau_{\mathcal{X}}) + \int_{\Theta} \int_{\mathcal{X}} \varphi(x, \theta) \, d(\nu - \nu_n)(x) \, d\gamma(\theta). \quad (5)$$

*Moreover, the solution $\nu^\star$ of (70) is precisely the Gibbs measure for the optimal $\gamma^\star$ in (3), that is, $\frac{d\nu^\star}{d\tau_{\mathcal{X}}}(x) = \frac{1}{Z_\beta} \exp\left( -\beta \int_{\Theta} \varphi(x, \theta) \, d\gamma^\star(x) \right)$.*

If we replace the $\mathcal{F}_1$ ball by the $\mathcal{F}_2$ ball, the analogous duality result links the maximum likelihood problem with the entropy regularized MMD flow from Arbel et al. (2019) (see App. C).

**Training dynamics on nonnegative measures.** Let us write the dynamics to solve (5) that can be discretized in terms of parameters and particles (cf Proposition 2 below). To this end we consider the triple $(\gamma^+, \gamma^-, \nu)$ where the nonegative measures $\gamma^\pm$ are defined through the Hahn decomposition of $\gamma = \gamma_+ - \gamma_-$. Then we introduce coupled gradient flows for this triple, in which $\gamma_t^+$ and $\gamma_t^-$ evolve via a Wasserstein-Fisher-Rao gradient flow (Chizat et al., 2018) and $\nu_t$ evolves via a Wasserstein gradient flow (Santambrogio, 2017):

$$\partial_t \gamma_t^\sigma = -\alpha\sigma\nabla_\theta \cdot \left(\gamma_t^\sigma \nabla_\theta F_t(\theta)\right) + \alpha\gamma_t^\sigma \left(\sigma F_t(\theta) - K_t\right), \qquad \sigma = \pm 1, \ \gamma_t^\sigma = \gamma_t^\pm$$

$$\partial_t \nu_t = \nabla_x \cdot \left(\nu_t \left(\nabla_x f_t(x) - \beta^{-1}\nabla\log\frac{d\tau_{\mathcal{X}}}{d\lambda}\right)\right) + \beta^{-1}\Delta_x\nu_t, \tag{6}$$

where $\alpha$ is a tunable parameter and we defined

$$F_t(\theta) = \int_{\mathcal{X}} \varphi(x, \theta)\, d(\nu_t - \nu_n)(x), \quad f_t(x) = \int_{\Theta} \varphi(x, \theta)\, (d\gamma_t^+ - d\gamma_t^-)(\theta), \tag{7}$$

$$K_t = \mathbb{1}_{\|\gamma_t^+\|_{\mathrm{TV}} + \|\gamma_t^-\|_{\mathrm{TV}} \geq 1} \int_{\Theta} F_t(\theta)(d\gamma_t^+ - d\gamma_t^-)(\theta).$$

The initialization of (6) is $\nu_0 = \nu_n$ and $\gamma_0^\pm = 0$ (such that the initial energy is null). The term $K_t$ keeps the total variation of $\gamma_t$ below one. The parameter $\alpha$ acts as a relative timescale. Notice that different values of $\alpha$ can potentially lead to different behaviors of the dynamics; setting $\alpha \ll 1$ would correspond to the primal formulation of maximum likelihood with persistent MCMC samples (as in PCD). In contrast if $\alpha \gg 1$, $\gamma_t^\pm$ evolves faster than $\nu_t$ and if the optimization is well behaved at all times $\gamma_t = \gamma_t^+ - \gamma_t^-$ remains close to minimizing the inner maximization problem of (5) with $\gamma = \gamma_t$. Initializing $\nu_0 = \nu_n$ is crucial to avoid the kind of metastabilities that curse the behavior of classical (primal) maximum likelihood EBM training.

Proposition 2 below states that the solution $(\mu_t, \nu_t)$ may be approximated using coupled particle systems (see proof in App. F) and is the basis for Alg. 1. The link between particle systems and measure PDEs is through a classical technique known as propagation of chaos (Sznitman, 1991) and it has been used previously for similar coupled systems in the machine learning literature (Domingo-Enrich et al., 2020).

**Proposition 2.** *Let $\{\theta_0^{(j)}\}_{j=1}^m$ be initial features sampled uniformly over $\Theta$, let $\{\sigma_j\}_{j=1}^m$ be uniform samples over $\{\pm 1\}$ and let $\{w_0^{(j)} = 1\}_{j=1}^m$ be the initial weight values, which are set to 1. Let $\{X_0^{(i)}\}_{i=1}^N$ be the initial "generated" samples, which are chosen i.i.d. uniformly from the target sample set $\{x_i\}_{i=1}^n$. Consider the system of ODEs/SDEs:*

$$\frac{d\theta_t^{(j)}}{dt} = \alpha\sigma_j\nabla\tilde{F}_t(\theta_t^{(j)}), \qquad \frac{dw_t^{(j)}}{dt} = \alpha w_t^{(j)}(\sigma_j\tilde{F}_t(\theta_t^{(j)}) - \tilde{K}_t)$$

$$dX_t^{(i)} = \left(-\nabla\tilde{f}_t(X_t^{(i)}) + \beta^{-1}\nabla\log\frac{d\tau_{\mathcal{X}}}{d\lambda}(X_t^{(i)})\right)dt + \sqrt{2\beta^{-1}}\, dW_t^{(i)} \tag{8}$$

*where*

$$\tilde{F}_t(\theta) = \frac{1}{N}\sum_{i=1}^N \varphi(X_t^{(i)}, \theta) - \frac{1}{n}\sum_{i=1}^n \varphi(x_i, \theta), \qquad \tilde{f}_t(x) = \frac{1}{m}\sum_{j=1}^m \sigma_j w_t^{(j)}\varphi(x, \theta_t^{(j)}),$$

$$\tilde{K}_t = \mathbb{1}_{\sum_{j=1}^m w_t^{(j)} \geq m} \frac{1}{m}\sum_{j=1}^m \sigma_j w_t^{(j)}\tilde{F}_t(\theta_t^{(j)}). \tag{9}$$

*are the empirical counterparts of the functions in (7). Then the system (8) approximates the measure dynamics. Namely, as $m, N \to \infty$:*

- *the empirical measure $\hat{\gamma}_t = \frac{1}{m}\sum_{j=1}^m \sigma_j w_t^{(j)}\delta_{\theta_t^{(j)}}$ converges weakly to the solution $\gamma_t = \gamma_t^+ - \gamma_t^-$ of (6) with uniform initialization for any finite time interval $[0, T]$, and*

- *the empirical measure $\hat{\nu}_t = \frac{1}{N}\sum_{i=1}^N \delta_{X_t^{(i)}}$ converges weakly to the solution $\nu_t$ of (6) for any finite time interval $[0, T]$.*

Importantly, the system of ODEs/SDEs in (8) may be solved via forward Euler steps on $\{\theta_j\}_{j=1}^m$ and $\{w_j\}_{j=1}^m$ (or rather, $\{\log w_j\}_{j=1}^m$), and Euler-Maruyama updates on $\{X^{(i)}\}_{i=1}^N$. Such a discretization yields Algorithm 1. Algorithm 1 makes use of a tunable parameter $p_R$, which stands for the restart probability and will be discussed in Sec. 4 as a natural way to connect maximum likelihood with score matching. To discretize (8) we set $p_R = 0$, i.e., there are no particle restarts.

---

**Algorithm 1** Dual $\mathcal{F}_1$-EBM training ($p_R = 0$: maximum likelihood, $p_R = (s\alpha \wedge 1)$: score matching)

**Input:** $n$ samples $\{x_i\}_{i=1}^n$ of the target distribution, stepsize $s$, stepsize ratio $\alpha$.
Initialize unif. features $(\theta_0^{(j)})_{j=1}^m$ over $\Theta$, weights $(w_0^{(j)})_{j=1}^m$ in $[0, 1)$, signs $(\sigma_j)_{j=1}^m$ over $\{\pm 1\}$.
Initialize generated samples $\{X_0^{(i)}\}_{i=1}^N$ uniformly i.i.d. from $\{x_i\}_{i=1}^n$.
**for** $t = 0, \dots, T-1$ **do**
  **for** $i = 1, \dots, N$ **do**
    With probability $p_R$, replace $X_t^{(i)}$ by some uniformly chosen sample in $\{x_i\}_{i=1}^n$ (see Sec. 4).
    Sample $\zeta_t^{(i)}$ from the $d_1$-variate standard Gaussian.
    Perform Euler-Maruyama update: $X_{t+1}^{(i)} = X_t^{(i)} - s(\nabla \tilde{f}_t(X_t^{(i)}) + \beta^{-1} \nabla \log \frac{d\tau_{\mathcal{X}}}{d\lambda}(X_t^{(i)})) + \sqrt{2\beta^{-1}s}\, \zeta_t^{(i)}$, where $\tilde{f}_t$ is defined in (9).
  **end for**
  **for** $j = 1, \dots, m$ **do**
    Update $\theta_{t+1}^{(j)} = \theta_t^{(j)} + s\alpha\sigma_j \nabla \tilde{F}_t(\theta_t^{(j)})$, where $\tilde{F}_t$ is defined in (9).
    Update $\tilde{w}_{t+1}^{(j)} = w_{t+1}^{(j)} \exp(s\alpha\sigma_j \tilde{F}_t(\theta_t^{(j)}))$.
    Normalize if needed $w_{t+1}^{(j)} = \tilde{w}_{t+1}^{(j)} / \max\left(m^{-1} \sum_{j'=1}^m \tilde{w}_{t+1}^{(j')}, 1\right)$.
  **end for**
**end for**
**Output:** samples $\{X_T^{(i)}\}_{i=1}^N$, energy $f_T(x) := \frac{\beta}{m} \sum_{j=1}^m \sigma_j w_j \varphi(x, \theta_j)$.

---

## 4   LINKS BETWEEN MAXIMUM LIKELIHOOD AND SCORE MATCHING $\mathcal{F}_1$-EBMS

In this section we uncover how the score matching loss fits seamlessly as a variant of Alg. 1, in the form of particle restarts. Interestingly, we can modify the PDE (6) in a way that allows us to make a connection with score matching. To this end, let us introduce the following coupled measure PDE:

$$\partial_t \gamma_t^\sigma = -\alpha\sigma\nabla_\theta \cdot \left(\gamma_t^\sigma \nabla_\theta F_t(\theta)\right) + \alpha\gamma_t^\sigma \left(\sigma F_t(\theta) - K_t\right), \qquad \sigma = \pm 1, \ \gamma_t^\sigma = \gamma_t^\pm,$$

$$\partial_t \nu_t = \nabla_x \cdot \left(\nu_t \left(\nabla_x f_t(x) - \beta^{-1} \nabla \log \frac{d\tau_{\mathcal{X}}}{d\lambda}\right)\right) + \beta^{-1} \Delta_x \nu_t - \alpha\left(\nu_t - \nu_n\right). \tag{10}$$

Remark that the only difference between this equation and the PDE (6) for dual maximum likelihood training is the term $-\alpha(\nu_t - \nu_n)$, which draws $\nu_t$ closer to the empirical target measure $\nu_n$. We have:

**Proposition 3.** *In the limit $\alpha \to \infty$, the equations for $\gamma_t^\sigma$ in (10) reduce to*

$$\partial_t \gamma_t^\sigma = \sigma\nabla_\theta \cdot \left(\gamma_t^\sigma \nabla_\theta V(\gamma_t)(\theta)\right) - \gamma_t^\sigma \left(\sigma V(\gamma_t)(\theta) - \bar{V}(\gamma_t)\right), \qquad \sigma = \pm 1, \ \gamma_t^\sigma = \gamma_t^\pm \tag{11}$$

*where $\gamma_t = \gamma_t^+ - \gamma_t^-$, $\bar{V}(\gamma) = \int_\Theta V(\gamma)d\gamma$, and $V(\gamma)(\theta)$ is the Frechet derivative of the score matching loss $L : \mathcal{M}(\Theta) \to \mathbb{R}$ defined in (4).*

That is, in the large $\alpha$ limit, equation (10) is equivalent to the Wasserstein-Fisher-Rao gradient flow of a loss $L$ which, remarkably, is the score matching loss for $\mathcal{F}_1$-EBMs. This means that adding the term $-\alpha(\nu_t - \nu_n)$ to the dual maximum likelihood measure dynamics and letting $\alpha \to \infty$ we recover the score matching dynamics. This additional term can be easily implemented at particle level by replacing each training sample $X_t^{(i)}$ by some random target sample in $\{x_i\}_{i=1}^n$ with probability $p_R = 1 - e^{-\alpha t} = \alpha t + o(t)$ for every time interval of length $t$ (proof in App. G).

Similar birth-death processes were used in (Rotskoff et al., 2019) in the context of neural network regression. Hence, the score matching scheme corresponds to setting the restart probability $p_R = s\alpha$ in Algorithm 1. The restart probability acts as a knob that allows us to interpolate between score matching and maximum likelihood.

In summary, score matching differs from dual maximum likelihood in that the trained measure is being "pulled" towards the target measure at all times via particle restarting. Such constant pulling should be useful to alleviate sampling problems due to metastability issues which may arise with dual maximum likelihood. However, dual maximum likelihood has the upside of providing samples of the learned EBM as a byproduct of training, which score matching does not. A good balance between both algorithms may be to use a restart probability $p_R$ between $s\alpha \wedge 1$ and 0, or even a $p_R$ that decreases with time from one value to the other, in such a way that at the beginning of training we avoid metastability issues by restarting the particles frequently, and at an advanced phase we perform little to no restarting to obtain faithful samples. It is also interesting to contrast our approach to score matching with the works Sutherland et al. (2018); Arbel & Gretton (2018), which using different techniques propose algorithms to train EBMs with RKHS energies via score matching. Finally, notice that a particle discretization of the flow (11) yields an alternative straightforward algorithm to train $\mathcal{F}_1$-EBMs via score matching; see Subsec. G.1. In Subsec. G.1 we show that this algorithm can be linked directly to Alg. 1 with particle restarts, without recurring to measure arguments.

## 5 EXPERIMENTS

**Setup.** To illustrate Alg. 1 we perform numerical experiments on simple synthetic datasets generated by teacher models with energy $f^*(x) = \frac{1}{J} \sum_{j=1}^{J} w_j^* \sigma(\langle \theta_j^*, x \rangle)$, with $\theta_j^* \in \mathbb{S}^d$ for all $j$. The training is performed using Alg. 1 with the added detail that both the features $\theta_t^{(j)}$ and the particles $X_t^{(i)}$ are constrained to remain on the sphere by adding a projection step in the update of their positions. The code, figures, and videos on the dynamics can be found in the supplementary material. In the main text we consider two planted teacher neurons ($J = 2$) with negative output weights $w_1^* = w_2^* = -10$ in dimension $d = 14$ and $m = 64$ neurons for the student model, but we include additional experiments and videos in App. I and supplementary material. We study setups with two different choices of angles between the teacher neurons, which showcase different behaviors:

- Teacher neurons $\theta_1^*, \theta_2^*$ forming an angle of 2.87 rad ($\approx 164$ degrees), and output weights $w_1^* = w_2^* = -10$. The teacher neurons are almost in opposite directions, and the resulting target distribution is bimodal, as the energy has two local minimizers around $\theta_1^*$ and $\theta_2^*$ (see Figure 5).

- Teacher neurons $\theta_1^*, \theta_2^*$ forming an angle of 1.37 rad ($\approx 78$ degrees). The teacher neurons are almost orthogonal and the resulting target distribution is monomodal; indeed, when the angle is less than $\pi/2$, the target energy has a unique minimizer at the geodesic average between $\theta_1^*$ and $\theta_2^*$ (see Figure 5 in App. I).

**Monitoring convergence.** In all our experiments, to monitor convergence we use a testing set of $n_*$ data points sampled from the teacher distribution: denoting these samples by $\{x_i^*\}_{i=1}^{n_*}$, we estimate the KL divergence from the student to the teacher via $\log(\frac{1}{n^*} \sum_{i=1}^{n^*} \exp(-\beta f_t(x_i^*) + \beta f^*(x_i^*))) + \frac{1}{n^*} \sum_{i=1}^{n^*} (f_t(x_i^*) + \beta f^*(x_i^*))$ where $f_t(x) = \frac{1}{m} \sum_{j=1}^{m} w_t^{(j)} \sigma(\langle \theta_t^{(j)}, x \rangle)$. Similarly, for the score matching objective we use the estimate $\frac{1}{n^*} \sum_{i=1}^{n^*} |\nabla_x f_t(x_i^*) - \nabla_x f^*(x_i^*)|^2$.

**Comparison of the primal algorithm and the dual algorithms.** We defer the empirical study of tuning the restart probability to App. I, and in this section focus on comparing the dual algorithm for maximum likelihood $\mathcal{F}_1$-EBMs (i.e. with $p_R = 0$) to the classical (primal) algorithm, which was the algorithm used in the experiments of Domingo-Enrich et al. (2021). The primal algorithm corresponds to Alg. 1 with $\alpha \ll 1$, while the dual algorithm uses $\alpha \gg 1$. To obtain a principled comparison of the two settings where numerical errors do not blow up, we set $s$ to be the step-size for the fastest process (particle evolution for the primal, neuron evolution for the dual), and $\min(\alpha, 1/\alpha)s$ the stepsize for the slow process.

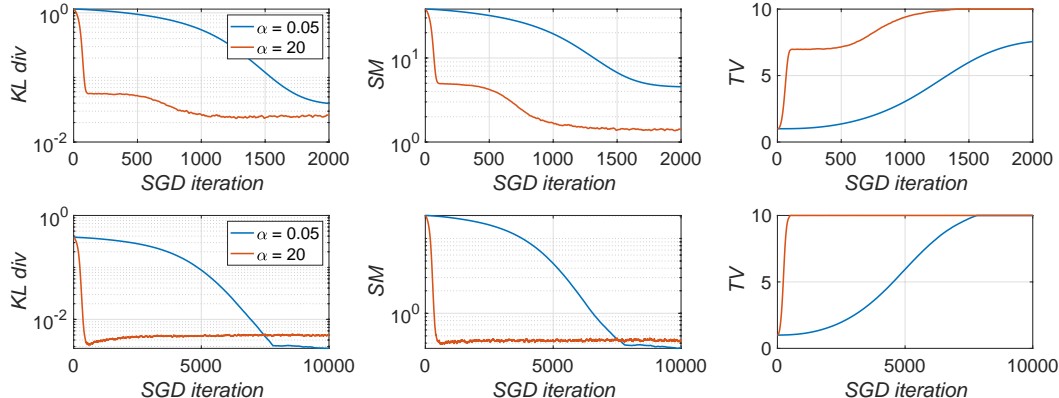

Figure 1: (Top) The evolution of the KL divergence, the score matching metric and the TV norm of the trained measure (i.e., the $\mathcal{F}_1$ norm) during training for Algorithm 1 with $\mathcal{X} = \mathbb{S}^{14}$, $m = 64$, $p_R = 0$, $s = 0.02$, $n = 10^5$, $N = 2 \cdot 10^5$ and $\alpha = 0.05$ (primal training) or $\alpha = 20$ (dual training), showing a speedup by a factor about 10-20 of the latter over the former. The angle between the two teacher neurons is 1.37 rad (monomodal distribution). (Bottom) Same experiments with an angle of 2.87 rad between the two teacher neurons (bimodal distribution).

The results are shown in Figure 6 for the two angle configurations between teacher neurons. We observe that the dual algorithm is several orders of magnitude faster at reaching KL and SM values close to the final ones, which showcases the main advantage of the dual approach. For the monomodal distribution, the final values obtained by the dual algorithm are slightly better than for the primal algorithm; for the bimodal distribution, the converse happens and the convergence is slower for both algorithms, most likely due to metastability. Interestingly, the decrease of the performance metrics seems to stall as soon as the hard $\mathcal{F}_1$-norm threshold is reached.

## 6 DISCUSSION AND OUTLOOK

In this work we leverage a Fenchel duality result to recast the maximum likelihood loss for $\mathcal{F}_1$-EBMs into a min-max problem on probability measures over the sample space. We provide mean-field dynamics at the measure level to solve this problem, which lead to a dual algorithm (Alg. 1) after discretization. We observe that if we restart particles at random target samples throughout training, we get an algorithm which is equivalent to training under the score matching loss. We perform experiments in which we learn planted distributions with two-layer ReLU networks, and we observe empirically that our dual algorithm is much faster than the classical one.

At theoretical level, one direction for future work is to obtain convergence results for the dynamics (6) and (10). Domingo-Enrich et al. (2020) study similar coupled Wasserstein-Fisher-Rao gradient flows, but their results only work in the case of weight learning rates much larger than position learning rates. We hypothesize that the additional term $-\alpha (\nu_t - \nu_n)$, which keeps $\nu_t$ close to $\nu_n$, might help in the analysis.

At the numerical level, it would be interesting to further test the variant of Algorithm 1 with annealed $p_R$ decreasing in time, to understand under which parameter setup it captures the best features of maximum likelihood and score matching. One could also test Algorithm 1 using deeper neural architectures: while the analysis is more complicated in this case, the scheme itself can be straightforwardly generalized to deep networks.

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
