# OpenReview forum: "Dual Training of Energy-Based Models with Overparametrized Shallow Neural Networks"
_ICLR.cc/2022/Conference — ICLR 2022 Submitted_

### Official Review · Reviewer_pG2E · 2021-10-28

**Correctness:** 4
**Technical Novelty And Significance:** 3
**Empirical Novelty And Significance:** 1
**Recommendation:** 5
**Confidence:** 4

**Main Review:**

The connection on a continuum of maximum likelihood and score based training is very interesting from a theoretical point of view, and I think this is a novel and valuable contribution.

My biggest concerns about the work are the following:

1) In the introduction it is stated that score matching is a weaker loss than maximum likelihood or, in other words, that score matching falls short in statistical power. However, the empirical evaluation in the paper shows that alpha>>1 (Score Matching) is superior to alpha<<1 (Maximum Likelihood). Isn’t this a contradiction? How do the authors explain this phenomenon? In summary I am missing a discussion along the following lines: “while Score Matching is weaker than Maximum Likelihood, our theoretical analysis shows that the impact on the convergence of alpha is …, and consequently we can explain why alpha >> 1 is better in practice”.

2) While I really appreciate the theoretical study in this paper, what I really miss is an implementation of the idea outlined at the end of section 4, which is what the authors refer to as an annealed p_R. This is exactly what the theoretical method needs: it enables (potentially) the development of a new algorithm that bounces from score matching to maximum likelihood, and achieves the best of both worlds. Such an algorithm could only have been conceived and theoretically justified with the work presented in section 3 and 4.
The “sentiment” I have is that the paper in its current form is not finished, because there is a disconnect between the theory and the empirical findings.

3) The experimental validation is somehow weak, as only a simple synthetic dataset is considered. If it wasn’t too complicated (as the authors suggest in their conclusion) to port Algorithm 1 to a deeper model (without a theoretical analysis, of course) why didn’t they do it?  Combining with point 1) above, it is hard to extrapolate any valid claims with a single and simple example. What guarantees do we have that the alpha>>1 would be better also in other settings? While it is not useful to explore deeper models, or for example begin with a comparison of the considered architecture (a 2 layer Neural Network) on standard datasets like UCI? This would clearly strengthen the paper.

Minor:
1) It is not clear why the discussion about the two different F_1 and F_2 spaces is important for the reader. Is there something more profound than “one of the two classes is more flexible than the other”? It could be useful for the reader to clarify this aspect, and in particular explain what is preventing the generalization of the result to deeper architectures. I am not sure I fully grasp where the model complexity coming into play in the theoretical development of the paper

2) Propagation of chaos is a technique I am not very familiar with, and I suspect it is not mainstream in the machine learning community. Providing some background on this would be helpful to produce a more self contained paper


**Summary Of The Paper:**

This paper studies the training of Energy Based Models (EBM). The “standard” technique is maximum likelihood maximization of the observed dataset. While theoretically sound, there are practical difficulties in simulating the MCMC dynamics due to large basins in the energy landscape. For this reason a class of alternative methods, like Score matching, have been explored in the literature. Score matching, however, is deemed to suffer in statistical power.

This work explores how, leveraging known results about Fenchel dualities, it is possible to connect the two training modalities (Maximum Likelihood and Score based) on a continuum, and proposes a practical algorithm based on such considerations.
The authors consider for their theoretical analysis of the expressiveness of the models,  shallow neural networks in the lazy and kernel regime (spaces F_1 and F_2 respectively).

Section 2.1 introduces the two considered spaces, referring to the works of Chizat and Bach.

An energy based model is defined at the beginning of section 2.2 by means of the Radon Nikodym derivative. Training an EBM model via maximum likelihood corresponds to maximizing eq. (2), that, when considering networks in F_1, can be rewritten as the expression (3). At the end of Section 2.2. score matching is rigorously defined and the relationship between theoretical and empirical implementations (Hyvarinen) is clarified.

Section 3 is the core of the paper. If Assumption 1 holds (growth conditions), then Thm 1 states that eq. (3) is the Fenchel dual of (5). As (5) is a static functional, the authors propose to use results from Chizat and Santambrogio to rewrite it as the dynamical system (eq. (6))  together with the definitions in (7). Importantly, this formulation contains \alpha, a free parameter >0, that determines relative time-scales over which functions \gamma and \nu are updated. System (6) is still “algorithmically” unsolvable. The authors then propose to use the two following approximations: first, the mean field technique known as propagation of chaos, is used to switch from (6)  to the system (8), where gamma is substituted by neural network  parameters and \nu by and SDE. Then the continuous time system (8) is discretized using Euler-Maruyama. Algorithm 1 is the result of the process.

Section 4 explores how, in the infinite alpha regime, the dynamical system (6) is equivalent to score matching. By means of eq. (11) and proposition (3) the technical analysis is performed. Intuitively, parameters are updated infinitely faster than the samples.

Section 5 explores the implementation of Algorithm 1 on a synthetic dataset. The authors consider two cases (the two teacher neurons aligned to 164 and 78 degrees respectively). Comparison of the primal and dual (alpha<<1, alpha>>1) shows that the dual performs much better than the primal.

The conclusions stress the theoretical contribution based on Fenchel duality and propose as future work the exploration of convergence rates of (6) and (10) and numerical investigation of the restarting probability p_R. In addition, the authors suggest that “porting” the presented work to deeper network architectures (e.g. to work on realistic dataset, such as images) is straightforward from the practical point of view, and unpractical from the theoretical point of view.


**Summary Of The Review:**

I think this work has a lot of potential, and the theoretical quest set by the authors is important.
However, the theoretical result per se is not fully exploited, and while the authors state this is on their research agenda, the current paper is weak on the algorithmic (the suggested idea of an annealed p_R should be developed) and empirical sides.

---

### Official Review · Reviewer_RMY3 · 2021-11-02

**Correctness:** 3
**Technical Novelty And Significance:** 3
**Empirical Novelty And Significance:** 2
**Recommendation:** 5
**Confidence:** 3

**Main Review:**

Let me start by saying that I strongly agree this energy based model is useful in many aspects of machine learning. But the key question I find myself asking is what the practical benefit of the proposed method is compared with scoring matching, and what difficulty it bypasses in the original MLE problem.

The author does not explain too much on why this dual problem is necessarily better than the primary problem other than saying the latter can be metastable. Below is my take: The MLE problem (3) is hard because of the difficulty of computing the partition function $Z_{\beta}$, while this task disappears in scoring matching. The dual problem (5) does not need to compute the partition function, but sill need to compute very similar high-D integral (the second term). At a very high level, (5) is akin to use adaptive-importances-sampling/bride sampling to compute $Z_{\beta}$, while we also optimize the proposal (base) density simultaneous on the fly.

Yet even it is much better than the primary problem, why is it better than the score matching (4), in which there is no need to compute  $Z_{\beta}$ at all? The experiment section is weak and there is no empirical evidence to support this claim. Yes, score matching can be derived as a special case by letting $\alpha \to \infty$, but score matching is much easier to train even without the proposed dynamics.

How is $\alpha$ tuned? How sensitive is the method to $\alpha$?

The structure of the paper is fine. But I am wondering is the introduction of $\mathcal{F}_1$ and  $\mathcal{F}_2$ totally relevant to the method development? Although I understand they imply two extremes of lazy and active learning, I don't find this difference is critical to the proposed method in the practical level.

eq 1 seems to be a typo.


**Summary Of The Paper:**

Two approaches exist for learning a Gibbs measure: either by MLE (minimizing the KL divergence) or by score matching (maximizing the Hyvarinen score). With the same goal to minimizing the KL divergence, the author proposes an alternative training approach by augmenting an auxiliary $\gamma$ measure and thereby optimizing a more tractable dual problem. The author claims that this new training method has a quicker convergence rate than the primary problem.

**Summary Of The Review:**

The proposed method is novel and promising for some useful applications. The paper is tightly presented although some reorganization can make the main contribution clearer. However, the theory justification, comparisons to the benchmark, and empirical evaluations are weak.

---

### Official Review · Reviewer_hfDM · 2021-11-08

**Correctness:** 3
**Technical Novelty And Significance:** 2
**Empirical Novelty And Significance:** 1
**Recommendation:** 3
**Confidence:** 3

**Main Review:**

This is a theoretically rich paper. I am not able to provide insightful technical comments in this urgent review, and I think the contributions in the main text and appendix are interesting though the significance is less well demonstrated. I personally find this paper hard to follow, with subtle unclear statements and jumps at various places.

Clarify:
1. What is meant by "active" and "lazy"? There are only two mentions of "active" and and three "lazy" in the combined main+appendix supplementary file. If these are stated in the abstract, they need to be clearly specified in the main.
2. The following statement:
> "Such models may be regarded as abstractions of more complex deep EBMs, in that they incorporate feature learning,
and they were first studied by Domingo-Enrich et al. (2021), which provide statistical guarantees.
What's meant by statistical guarantees? Statements like this make readers second guess what the authors want to convey. [The ref of this highly related paper indicates that the paper is not yet published, but it is actually accepted in May and published in July 2021, well before the current submission deadline.]
3. There is a huge jump from (5) to (6), which I cannot follow, with only 8 lines of text explanations. What's $K_t$?
4. The remark at the end of page 3: what's $L1(Θ)$ norm? Is this related to the TV norm?

Experiments:
The experiment demonstrates the simplest possible setting, which I reckon is of little interest to most readers of this conference. Most other cited and theoretically rich papers demonstrate at least on a few real datasets. In particular, no experiments try to verify the Proposition 3, one of the more interesting theoretical results in my opinion. This makes it hard to support this paper in terms of its significance.

Minor issue:
For the state results on score matching to hold, the definition of the score matching (relative Fisher information) should be
defined with a factor of 1/2 in the proof to Proposition 1.

**Summary Of The Paper:**

The authors formulated the Fenchel dual problems to maximum likelihood and score matching training applied to energy-based models defined by shallow neural networks. The authors then proposed practical algorithms based on the Hahn decomposition and compared this to various predecessor algorithms. An interesting part is that the learning dynamics can interpolate between maximum likelihood and score matching. There are also rich results in the Appendix.

**Summary Of The Review:**

This paper potentially has rich and interesting theoretical insights, but the results are very poorly described and insufficiently demonstrated in experiments.

---

### Official Review · Reviewer_aaPp · 2021-11-08

**Correctness:** 4
**Technical Novelty And Significance:** 3
**Empirical Novelty And Significance:** 2
**Recommendation:** 6
**Confidence:** 3

**Main Review:**

Strengths:
* As far as I know, the paper is self-contained and technically sound. The author introduces the motivation and derivation in detail. The proposed algorithm naturally connects MLE and SM.
* Numerical experiments seem to show that it is a promising alternative to train EBMs.

Weaknesses:
* I would say that the notation and formulation are a bit heavy and not familiar to the ML community, which makes the paper really hard to follow. I have to spend quite a long time reading through the paper and getting the points.
* The paper empirically shows that the dual algorithm trains the EBM faster. However, I would like to see more theoretical analysis in terms of the convergence speed.
* The numerical experiments are all based on synthetic datasets generated by predefined EBMs. I would like to see at least one example of applying the proposed algorithm to a realistic dataset (e.g., MNIST).

**Summary Of The Paper:**

This paper derives a Fenchel duality formulation of the maximum likelihood loss of F1-EBMs, which turns the optimization into a min-max problem on probability measures over the sample space. A dual algorithm with mean-field dynamics is proposed to solve this problem. Using the dual aogrithm they also draw a connection between maximum-likelihood training and score matching. Simple numerical experiments on two-layer ReLU network show that the algorithm converges much faster than the original MLE.

**Summary Of The Review:**

Overall speaking, this is a decent and novel paper analyzing the training algorithm of EBMs from the perspective of probability measure. The paper could be strengthened if more theoretical analysis of convergence speed and more empirical results are included.

---

### Author Response · Authors · 2021-11-23
**General response**

We would like to thank the reviewers for their work. Their comments will be useful for us to improve our paper.

---

### Decision · Program_Chairs · 2022-01-20

**Decision:**

Reject

**Comment:**

In this paper, the authors generalized the Fenchel duality formulation of the maximum likelihood for F1-EBM, which leads to a min-max optimization formulation. Meanwhile, the optimization reveals a new connection between primal-dual MLE with score matching. These contribution is significant and make the paper interesting to the community.

However, there are several issues need to be addressed.

- *Experiments*: most of the reviewers concern about the empirical study, which are conducted on synthetic data. The paper will be much stronger if the comparison on real-world dataset, e.g., MNIST and CIFAR10, can be conducted.

- *Clarification of paper*: I totally understand that due to this is a theoretical-oriented paper, it must be notation and derivation heavy. However, the paper will be much easier for reader, if more discussion is added, e.g., the implementation of the proposed algorithms and more explanation about the comparison between primal-dual algorithm vs. score matching and the experimental results for broader audiences.


In fact, I personally like the paper very much, which provides a promising solid algorithm for EBM estimation, and the connection to score matching is also novel and different from the current understanding. However, unfortunately, the authors gave up the rebuttal and did not successfully address the concerns from the reviewers. I strongly encourage the authors to revise the draft according to the reviewers' suggestion and resubmit to another venue.